# Parameter estimation for nonlinear sandwich system using instantaneous performance principle

**Zhengbin Li**[1]*, **Lijun Ma**[2], **Yongqiang Wang**[3]

**1** School of electronic information and electrical engineering, Anyang Institute of Technology, Anyang, Henan, China, **2** Power plant, Anyang Iron & Steel Co., Ltd., Anyang, Henan, China, **3** Iron-making Plant, Anyang Iron & Steel Co., Ltd., Anyang, Henan, China

☯ These authors contributed equally to this work.
\* ligzwq08052008@163.com

**Data Availability Statement:** All relevant data are within the paper and its Supporting information files.

**Funding:** This paper is supported by the Key Specialized Research and Development Projects of

## Abstract

The vast majority of reports mainly focus on the steady-state performance of parameter estimation. Few findings are reported for the instantaneous performance of parameter estimation because the instantaneous performance is difficult to quantify by using the design algorithm, for example, in the initial stage of parameter estimation, the error of parameter estimation varies in a specific region on the basis of the user's request. With that in mind, we design an identification algorithm to address the transient performance of the parameter estimations. In this study, the parameter estimation of nonlinear sandwich system is studied by using the predefined constraint technology and high-effective filter. To achieve the above purpose, the estimation error information reflecting the transient performance of parameter estimation is procured using the developed some intermediate variables. Then, a predefined constraint function is used to prescribe the error convergence boundary, in which the convergence rate is lifted. An error equivalent conversion technique is then employed to obtain the transformed error data for establishing an parameter adaptive update law, in which the estimation error convergence and the predefined domain can be achieved. In comparison with the available estimation schemes, the good instantaneous performance is obtained on the basis of the numerical example and practical process results.

## 1. Introduction

System identification technology in automatic control, signal processing, model prediction, and fault diagnosis has become interesting owing to the advancement in data acquisition and high-accuracy in the model obtained [1]. In system identification, the parameter estimation based on a specific nonlinear modeling technique is widely investigated. Sandwich model is a common nonlinear modeling method, where the linear and nonlinear parts are interconnected [2, 3], as shown in Fig 1. Sandwich model can effectively represent a mathematical dynamic model of actual nonlinear system including electromechanical system, optical transmitter, and bipolar electrosurgery, etc. [4, 5]. Hence, the parameter estimation of sandwich system is helpful to understand the dynamic characteristics of actual processes.

Henan Province under Grant 202102210337. The funder provided support in the form of salaries for authors [Z. Li], but did not have any additional role in the study design, data collection and analysis, decision to publish, or preparation of the manuscript. The specific roles of these authors are articulated in the 'author contributions' section.

Numerous identification methods have been proposed to handle the parameter estimation of the sandwich system [6–8]. In Shaikh *et al*. [9], a spearman correlation scheme is used to identify the sandwich model, in which the good initial values are obtained by using the best linear approximation approach. Liu [10] proposed an improved bayesian approach to calculate the posterior distribution of the internal variables, used expectation maximization scheme to produce the estimation values of the sandwich system parameters. Dreesen *et al* [11] used canonical polyadic decomposition to decompose Volterra model into a sandwich system, and applied least squares to estimate system parameters. With the help of the auxiliary model, a multi-innovation gradient method is proposed by Xu [12] to address the parameter estimation of sandwich system. Li *et al* [6] reported an adaptive estimator for the considered system where the adaptive law is designed through the usage of the parameter error and initial value. An efficient gradient estimation method is given in [13] by Campo *et al*. for the sandwich model, and the model is used to build a dynamic model of a nonlinear radio system. Although the aforementioned-published identification algorithms have achieved good results in sandwich system identification, these algorithms focus on the steady-state performance of sandwich system parameter identification, i.e., $t \rightarrow \infty, \tilde{\theta}(t) \rightarrow \theta$. Few reports on the transient performance of parameter estimation are published. This can be responsible for the fact that because the instantaneous performance is difficult to quantify by using the design algorithm. As a matter of fact, it retains as a provocative and open task to quantitatively assess the estimation error instantaneous performance before realizing steady-state performance. Another reason to consider parameter estimation transient performance is that fast and good transient performance can contribute to online real-time adjustment of control parameters and thus improve the control performance [14, 15]. Therefore, it is an interesting and necessary problem to discuss the transient performance of parameter estimation.

To enhance the estimation accuracy and address the biased estimates problem, the filter was proposed to achieve the parameter estimation and system identification communities [16, 17]. A polynomial filtering technique was reported in [18] to filter the input and output data, and a partially-coupled stochastic gradient method was proposed to conduct parameter estimation. In [19], the parameter identification of a nonlinear system was studied by using an auxiliary filter, the convergence speed was improved. Using a Kalman filter, an optimal Bayesian identification scheme was developed for nonlinear state-space models, where state and parameter estimation were implemented simultaneously [20]. In the aforementioned parameter identification approaches based on the filters, the filter performance is achieved based on the several assumptions such as positive definite condition [18], many adjustment parameters [21], and prior knowledge [22], etc. To relieve the assumptions, a simple robust unknown identification algorithm with linear filter was proposed to identify lumped parameter of nonlinear systems, in which the exponential error convergence was reached [23, 24]. However, although the structure of filter is simple, but the assumptions of the filter need be further relaxed. Therefore, the current work is dedicated to develop a filter with less assumption.

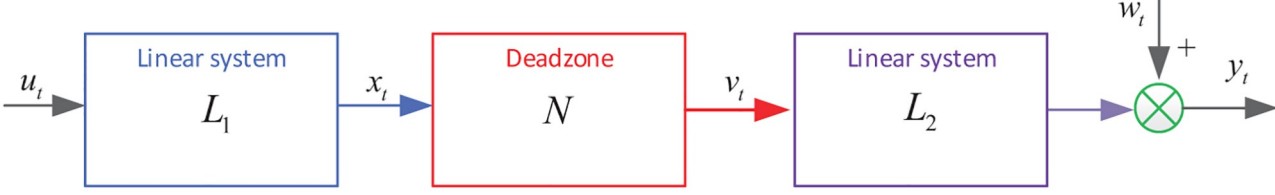

**Fig 1. Nonlinear sandwich system.**

In this study, motivated by above discussions, an instantaneous performance scheme of the parameter estimation is proposed to estimate sandwich system parameters, which can achieve the instantaneous performance of the estimation error convergence in the first few stage, apart from the ensured steady state performance. To this end, an improved predefined constraint technology (IM-PCT) is proposed to set error convergence boundary of the parameter estimation. By introducing a novel filter and filtered variables, the estimation error information is extracted, then an error equivalent conversion technique is further used to derive the parameter adaptive law on the basis of some forcing variables. Finally, the superiority of the proposed algorithm are tested by force of numerical simulation and experimental platform.

The main contributions of this study are listed as follows:

(1) A filter gain is developed by considering simple form, such that the assumptions can be further relaxed [25, 26].

(2) By designing several forcing variables with variable fading factor, the identification error data can be procured which can reflect the instantaneous performance of the estimator, and gives a solution to address the instantaneous behaviour of parameter identification in system identification communities.

(3) A novel framework of identification scheme is provided by containing identification error data and IM-PCT, so that the instantaneous nature of the identifier is predefined based on the requirement of users compared with the conventional identification algorithm [27, 28].

The rest of this paper is summarized as follows. In Section 2, sandwich system and identification model are offered. Section 3 provides the presented identification scheme. In Section 4, the theoretical analysis of the designed method in Section 3 is reported. The verification results on the example and experiment are described in Section 5. Finally, the conclusions are listed in Section 6.

## 2. Sandwich system and identification model

This section introduces the considered sandwich system and identification model. In Fig 1, the sandwich system consists of two linear systems $L_1$ and $L_2$, a nonlinear deadzone model $N$. The linear systems are described by:

$$L_1 : x_t = \sum_{j=1}^{n_a} a_j u_{t-j} - \sum_{i=1}^{n_b} b_i x_{t-i}, \tag{1}$$

$$L_2 : y_t = \sum_{j=1}^{n_c} c_j v_{t-j} - \sum_{i=1}^{n_d} d_i y_{t-i}. \tag{2}$$

The middle deadzone is expressed by the following piecewise expression:

$$N(\cdot) : v_t = \begin{cases} k_L(x_t + d_L), & if \ x_t \leq -d_L \\ 0, & if -d_L < x_t < d_R \\ k_R(x_t - d_R), & if \ x_t \geq d_R, \end{cases} \tag{3}$$

where the former linear subsystem is denoted by $L_1$, the latter linear subsystem is described by $L_2$, the middle nonlinearity is represented by $N(\cdot)$. $k_L$, $k_R$ are slopes of deadzone, $d_L$, $d_R$ describe the two end-points. $u_t$, $y_t$ denote the input-output of the system, $w_t$ is addition noise, $x_t$ is the output of $L_1$, the output of $N(\cdot)$ is denoted by $v_t$.

**Assumption 1**. *(I) Parameter uniqueness condition: the first coefficient of $L_1$ and the first coefficient of $L_2$ are set to one, i.e., $a_1 = 1$ and $c_1 = 1$. (II) Persistent excitation condition: when the input signal $u_t$ is a continuous excitation signal, all modes of the system can be excited. (III) The degrees information of two linear systems are assumed to be known.*

Assumption (I) shows the parameter uniqueness condition. In assumption (II), the system is excited by using the chosen input signal. Assumption (III) displays that the linear system orders are known. These assumptions can be found in [29, 30].

In order to reduce the estimated parameters redundancy, the separation theory of key-term [31] is used to address the identification model. By inserting (1), (3) into (2) and combining the separation theory of key-term, the compact form of identification model is given by

$$y_t = \theta^T \varphi_t + w_t, \tag{4}$$

where system data $\varphi_t$ is expressed by

$$\varphi_t = [h_{1,t-1}u_{t-2}, h_{1,t-1}u_{t-3}, \cdots, h_{1,t-1}u_{t-1-n_a}, -h_{1,t-1}x_{t-2}, \cdots, -h_{1,t-1}x_{t-1-n_b}$$
$$, h_{1,t-1}, h_{2,t-1}x_{t-1}, -h_{2,t-1}, v_{t-2}, v_{t-3}, \cdots, v_{t-n_c}, -y_{t-1}, -y_{t-2}, \cdots, -y_{k-n_d}]^T, \tag{5}$$

and the estimated parameter $\theta$ is written as

$$\theta = [k_L c_1 a_1, k_L c_1 a_2, \cdots, k_L c_1 a_{n_a}, k_L c_1 b_1, \cdots, k_L c_1 b_{n_b}, k_L d_L c_1, k_R c_1, k_R d_R c_1$$
$$, c_2, \cdots, c_{n_c}, d_1, d_2, \cdots, d_{n_d}]^T, \tag{6}$$

where deadzone linearization functions are described by

$$s[t] = \begin{cases} 1, & if \ t \leq 0 \\ 0, & if \ t > 0, \end{cases} \tag{7}$$

$$h_{1,t} = s[x_t - d_L],$$
$$h_{2,t} = s[d_R - x_t]. \tag{8}$$

This paper aims at estimating the system unknown parameters $a_j$, $k_L$, $d_L$, $b_i$, $k_R$, $c_j$, $d_R$ and $d_i$ by proposing a robust instantaneous performance identification method for the sandwich system, analyzing the convergence of the presented approach by force of the martingale theorem, testing the usefulness and practicality of the proposed algorithm in this paper.

## 3. Robust instantaneous performance estimator

We present a novel instantaneous performance identification scheme to identify the sandwich system in this section. Different from conventional identification algorithm, an instantaneous performance estimator rather than the steady-state performance estimator is proposed, which gives a new framework of estimator based on identification error and IM-PCT technique, so that the instantaneous performance can be realized. The diagram of the developed estimator is displayed in Fig 2.

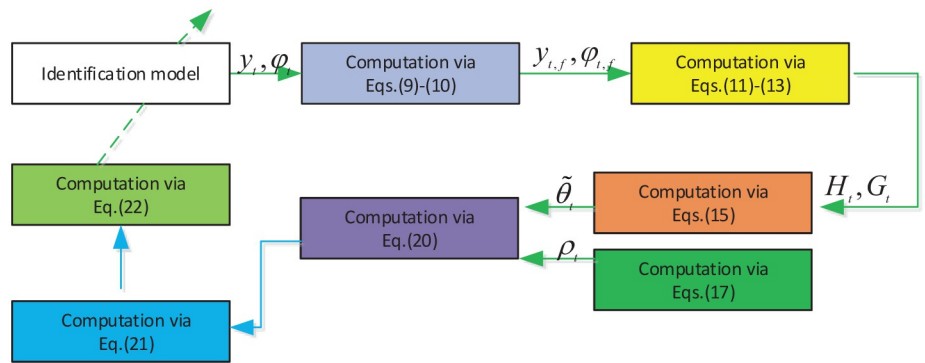

**Fig 2. Schematic of sandwich system with proposed scheme.**

As shown in (4)–(6), the output $y_t$ and data vector $\varphi_t$ involve the noise data. Based on this, $y_t$ and $\varphi_t$ are filtered by proposing a filter $v$. The filtered variables $y_{t,f}$ and $\varphi_{t,f}$ are provided by

$$y_{t,f} = \frac{v}{v+1} y_{t-1,f} + \frac{1}{v+1} y_t, \tag{9}$$

$$\varphi_{t,f} = \frac{v}{v+1} \varphi_{t-1,f} + \frac{1}{v+1} \varphi_t, \tag{10}$$

where $v$ is the developed filter.

As stated earlier, a variable that can represent the instantaneous performance of the estimator is needed when PCT is applied. Based on the definition of the identification error, we know that it can reflect the instantaneous performance of parameter identification. Unfortunately, the error information of the identification is unknown in the estimation process. In order to solve this obstacle, only using the collected system data, we need to design a method to obtain the identification error information. In this study, we will propose a solution to derive the identification error variable through the usage of some intermediate variables $H_t$, $G_t$ and $V_t$. Now, the derivation process is as follows:

With the help of (9)-(10), the matrix $H_t$ and vector $G_t$ can be written as follows:

$$H_t = \frac{1}{1+\kappa_t} H_{t-1} + \frac{1}{1+\kappa_t} \varphi_{t,f} \varphi_{t,f}^T, \tag{11}$$

$$G_t = \frac{1}{1+\kappa_t} y_{t,fl} \varphi_{t,fl}^{-1} H_{t-1} + \frac{1}{1+\kappa_t} y_{t,f} \varphi_{t,f}^T, \tag{12}$$

$$\kappa_t = \epsilon e^{-\varsigma t}/(1+e^{-\varsigma t})^2, \tag{13}$$

where the variable fading factor is represented by $\kappa_t$. $\varsigma, \epsilon > 0$. $y_{t,fl} = y_g \varphi_g^T$, $\varphi_{t,fl} = \varphi_g \varphi_g^T$, $y_g = [y_{1,f}, \cdots, y_{M,f}]$, $\varphi_g = [\varphi_{1,f}, \cdots, \varphi_{M,f}]$, $M > 0$. The initial values of $H_t$, $G_t$ are chosen as small values.

**Remark 1**. *The variable fading factor $\kappa_t$ in (13) is used to lift the utilization of the "new" system data other than manually tuned constant forgetting factor, which can improve the so-called data submerge problem of identification community.*

Go a step further, based on (11)–(13), the vector $V_t$ is defined by the following form:

$$V_t = \hat{\theta}_{t-1}^T H_t - G_t + \upsilon_t,  \qquad (14)$$

where $\upsilon_t = 1/(1 + \kappa_t) w_{t,f} \varphi_{t,f}^T$, $w_{t,f}$ is the filtered value for $w_t$.

By inserting (11)-(12) into (14), $V_t$ can be written as

$$\begin{aligned} V_t &= \hat{\theta}_{t-1}^T H_t - G_t + \upsilon_t \\ &= \hat{\theta}_{t-1}^T H_t - \frac{1}{1+\rho} y_{t,fl} \varphi_{t,fl}^{-1} H_{t-1} - \frac{1}{1+\rho} y_{t,f} \varphi_{t,f}^T + \upsilon_t \\ &= -\tilde{\theta}_{t-1}^T H_t, \end{aligned}  \qquad (15)$$

or

$$\tilde{\theta}_{t-1} = \left(-V_t H_t^{-1}\right)^T,  \qquad (16)$$

where the identification error $\tilde{\theta}_t$ is defined by $\tilde{\theta}_t = \theta - \hat{\theta}_t$.

At this point, the identification error data $\tilde{\theta}_t$ is obtained by using (16). Based on $\tilde{\theta}_t$, PCT technique can be used to improve the instantaneous performance of the estimator. A decreasing function $\varrho_t$ is set to the PCT as the works [32]. The following improved PCT (IM-PCT) is used to enhance the convergence rate of the PCT in this paper:

$$\varrho_t = \varrho_0 e^{-\gamma t} + \varrho_\infty \frac{t}{\gamma t + \gamma},  \qquad (17)$$

where $0 < \varrho_\infty < \varrho_0 < \infty$, $\lim\limits_{t \to \infty} \varrho_t = \varrho_\infty$, $\gamma \geq 1$,

**Remark 2**. *As displayed in* Fig 3, *IM-PCT possesses less time-consuming at the beginning of parameter estimation, which indicates that it provides faster convergence speed because the decay rate of $1/x$ proposed by this paper is higher than that of $1 - e^{-x}$ given by classic PCT* [33, 34].

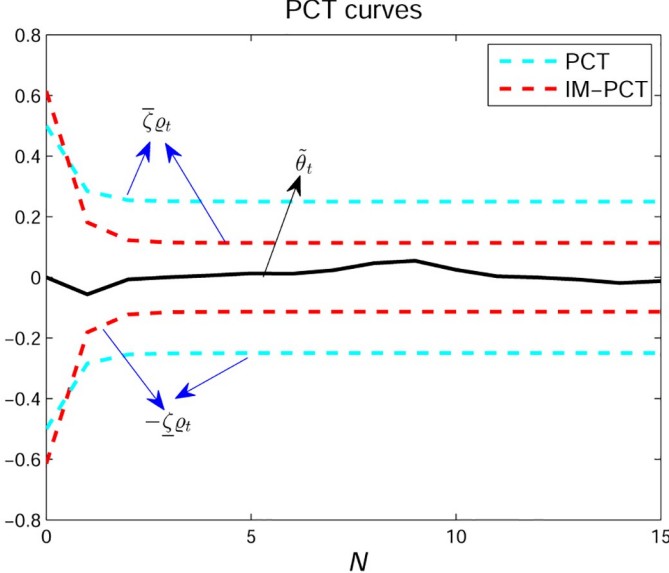

**Fig 3. PCT results.**

By the aid of (17), the identification error can be expressed by

$$-\underline{\zeta}\varrho_t < \tilde{\theta}_t < \bar{\zeta}\varrho_t, \tag{18}$$

where $\underline{\zeta}, \bar{\zeta} > 0$.

**Remark 3**. *As shown in* (18), *when* $\varrho_t = \varrho_0$, *the lower and upper bounds of error overshoot are* $-\underline{\zeta}\varrho_0$ *and* $\bar{\zeta}\varrho_0$, *respectively, as shown in* Fig 3. *When the system tends to steady state,* $\varrho_\infty/\gamma$ *describes the upper value of the error. The convergence rate is represented by* $\gamma$. *Thus, by designing* $\varrho_0, \bar{\zeta}, \gamma, \underline{\zeta}$ *and* $\varrho_\infty$, *the developed approach can give a better instantaneous property.*

In order to content the error constrained condition (18), a strictly increasing function $W(\Theta_t), \lim_{\Theta_t \to \infty} W(\Theta_t) = \bar{\zeta}, \lim_{\Theta_t \to -\infty} W(\Theta_t) = -\underline{\zeta}$ is chosen as follows [35]:

$$W(\Theta_t) = \frac{\bar{\zeta} - \underline{\zeta}e^{-\Theta_t}}{1 + e^{-\Theta_t}}. \tag{19}$$

By using (19), (18) can be rewritten as follows:

$$\tilde{\theta}_t = \varrho_t W(\Theta_t), \tag{20}$$

where the transformed error is denoted by $\Theta_t$.

**Remark 4**. [36] *In accordance with the identification error* $\tilde{\theta}_t$, *the condition in* (18) *and transformed error* $\Theta_t$ *given in* (20), *when the convergence of* $\Theta_t$ *is achieved, the predefined performance of* $\tilde{\theta}_t$ *can be satisfied, that is, the condition in* (18) *is realized.*

To achieve the predefined performance of $\tilde{\theta}_t$, we develop the following recursive form of $\hat{\Theta}_t$:

$$\hat{\Theta}_t = \hat{\Theta}_{t-1} + \chi \left[ \hat{\theta}_t - \hat{\theta}_{t-1} - \tilde{\theta}_{t-1}\left(1 - \frac{\varrho_{t-1}}{\varrho_t}\right) \right], \tag{21}$$

$\chi = [\underline{\zeta} + \bar{\zeta}]/[(\tilde{\theta}_t + \underline{\zeta}\varrho_t)(\varrho_t\bar{\zeta} - \tilde{\theta}_t)], 0 < \chi_{mim} < \chi < \chi_{max} < \infty$.

As Remark. 4 states, the realization of the preset performance depends on the convergence of $\Theta_t$. With that in mind, the issue now is to design an adaptive learning law for $\hat{\theta}_t$, to ensure the convergence of $\Theta_t$.

The parameter adaptive learning law for $\hat{\theta}_t$ is defined by:

$$\hat{\theta}_t = \hat{\theta}_{t-1} - \Gamma_t H_t \frac{\Delta_{t-1}}{\|\Delta_{t-1}\|} - [V_t H_t^{-1}]^T \left(1 - \frac{\varrho_{t-1}}{\varrho_t}\right), \tag{22}$$

where $\Gamma_t$ represents modified gain, $\Delta_{t-1} = \hat{\Theta}_{t-1} - \ln(\underline{\zeta}/\bar{\zeta}) + v_t = -\tilde{\Theta}_{t-1} + v_t$, $\tilde{\Theta}_t$ is the auxiliary error of $\Theta_t$.

To deal with variables $x_t$ and $v_t$ that are not measured directly, the reference model idea [37, 38] is applied to obtain indirect value. The basic idea is to displace $x_t$ and $v_t$ using the outputs $x_{t,ref}$ and $v_{t,ref}$ of auxiliary model (see Fig 4). $x_{t,ref}$ and $v_{t,ref}$ are reconstructed by:

$$x_{t,ref} = \sum_{j=1}^{n_a} \hat{a}_j u_{t-j} - \sum_{i=1}^{n_b} \hat{b}_i x_{t-i,ref}, \tag{23}$$

$$v_{t,ref} = \hat{k}_L x_{t,ref} \hat{h}_{1,t} + \widehat{k_L d_L} \hat{h}_{1,t} + \hat{k}_R x_{t,ref} \hat{h}_{2,t} - \widehat{k_R d_R} \hat{h}_{2,t}. \tag{24}$$

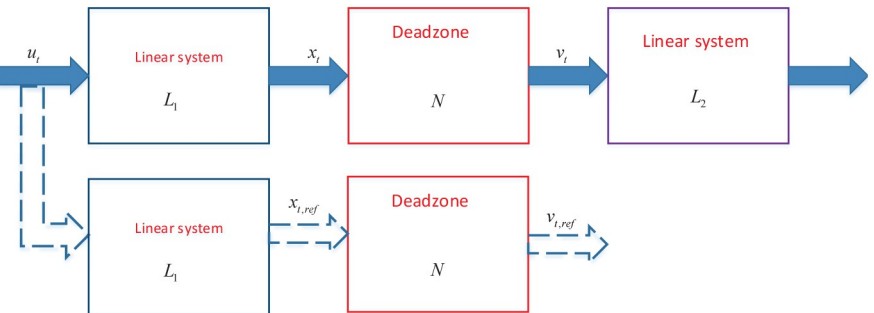

**Fig 4. Auxiliary model structure.**

**Remark 5**. *By substituting $x_t$ and $v_t$ in $\varphi_t$ using $x_{t,ref}$ and $v_{t,ref}$, then $\varphi_t$ can be measured indirectly. Now, $x_t$, $v_t$ and $\varphi_t$ are known indirectly, the other variables that contain $x_t$, $v_t$ and $\varphi_t$ are also known. Then, the proposed identification method* Eqs (9)–(24) *is conducted, such that the instantaneous performance can be achieved.*

For the parameters $v(0)$, $\varrho_\infty$, $\underline{\zeta}$ and $\overline{\zeta}$, which are dependent on user in general. However, when the parameters are chosen, we can refer to the initial system condition information. To obtain good transient performance and based on $0 < v(0) < 1$, $0 < \varrho_\infty < 1$, $0 < \underline{\zeta} < 1$ and $0 < \overline{\zeta} < 1$, we can limit the difference between the estimated value and the initial value to 5% -50%, so that the overshoot will not be too large.

## 4. Convergence analysis

In this section, the convergence quality of the developed scheme in Section 3 is analyzed using the martingale theorem.

Assume that $\{v_t, \mathscr{F}_t\}$ is a bounded martingale, in which $\sigma$ algebra sequence $\{\mathscr{F}_t\}$ is constituted by $\{v_t\}$, and the noise $\{v_t\}$ satisfies [39]:

(C1) $E[v_t|\mathscr{F}_{t-1}] = 0$, *a.s.*,

(C2) $E[\|v_t\|^2|\mathscr{F}_{t-1}] = \sigma_{t,v}^2 \leq \sigma_v^2 < \infty$, *a.s.*,

(C3) $\limsup\limits_{t\to\infty} \dfrac{1}{t} \sum\limits_{j=1}^{t} \|v_j\|^2 \leq \sigma_v^2 < \infty$, *a.s.*.

**Theorem 1**. *For the model* (4) *and the proposed algorithm* (9)–(24), *suppose that the excitation signal $u_t$ satisfies persistent excitation condition, such that $\sum\limits_{j=1}^{t} \varphi_{j,f}\varphi_{j,f}^T > \mu I, \mu > 0$ and (C1)-(C3) hold.*

*Then, when the following error bound is guaranteed for $\tilde{\Theta}_t$:*

$$\|\tilde{\Theta}_t\|^2 \leq O\left(\frac{[\ln|H_t|]^\epsilon}{\lambda_{min}[\Gamma_\Theta^{-1}]}\right),$$

*the predefined performance* (18) *in Section 3 is realized.*

Now, the proof of Theorem.1 is shown as follows:

*Proof.* By inserting (22) into (21), we have

$$\hat{\Theta}_t = \hat{\Theta}_{t-1} - \chi \Gamma_{t,\Theta} H_t \frac{\Delta_{t-1}}{\|\Delta_{t-1}\|}. \tag{25}$$

Firstly, we define function $S_t = \tilde{\Theta}_t^T \Gamma_{t,\Theta}^{-1} \tilde{\Theta}_t$. Based on (25), it yields

$$
\begin{aligned}
\ln(\underline{\zeta}/\bar{\zeta}) - \hat{\Theta}_t &= \ln(\underline{\zeta}/\bar{\zeta}) - \hat{\Theta}_{t-1} + \chi \Gamma_{t,\Theta} H_t \frac{\Delta_{t-1}}{\|\Delta_{t-1}\|} \\
\tilde{\Theta}_t &= \tilde{\Theta}_{t-1} + \chi \Gamma_{t,\Theta} H_t \frac{\Delta_{t-1}}{\|\Delta_{t-1}\|}.
\end{aligned}
\tag{26}
$$

By substituting (26) into $S_t$, we obtain

$$
\begin{aligned}
S_t &= \tilde{\Theta}_t^T \Gamma_{t,\Theta}^{-1} \tilde{\Theta}_t \\
&= \left[ \tilde{\Theta}_{t-1} + \chi \Gamma_{t,\Theta} H_t \frac{\Delta_{t-1}}{\|\Delta_{t-1}\|} \right]^T \Gamma_{t,\Theta}^{-1} \left[ \tilde{\Theta}_{t-1} + \chi \Gamma_{t,\Theta} H_t \frac{\Delta_{t-1}}{\|\Delta_{t-1}\|} \right] \\
&= \tilde{\Theta}_{t-1}^T \Gamma_{t,\Theta}^{-1} \tilde{\Theta}_{t-1} + 2\chi \tilde{\Theta}_{t-1}^T H_t \frac{\Delta_{t-1}}{\|\Delta_{t-1}\|} + \chi^2 \frac{\Delta_{t-1}^T H_t \Gamma_{t,\Theta} H_t \Delta_{t-1}}{\|\Delta_{t-1}\|^2} \\
&= S_{t-1} - 2\chi \frac{\tilde{\Theta}_{t-1}^T H_t \tilde{\Theta}_{t-1}}{\|\Delta_{t-1}\|} + 2\chi \frac{\tilde{H}_t \upsilon_t}{\|\Delta_{t-1}\|} + \chi^2 \frac{\Delta_{t-1}^T H_t \Gamma_{t,\Theta} H_t \Delta_{t-1}}{\|\Delta_{t-1}\|^2} \\
&\leq S_{t-1} + 2\chi \frac{\tilde{H}_t \upsilon_t}{\|\Delta_{t-1}\|} + \chi^2 \lambda_{max}[H_t \Gamma_{t,\Theta} H_t],
\end{aligned}
\tag{27}
$$

where $\tilde{H}_t = \tilde{\Theta}_{t-1}^T H_t$, $2\chi \tilde{\Theta}_{t-1}^T H_t \tilde{\Theta}_{t-1}/\|\Delta_{t-1}\| > 0$.

Because $\chi^2 \lambda_{max}[H_t \Gamma_{t,\Theta} H_t]$ and $\tilde{H}_t$ are not related to $\upsilon_t$, for (27), based on the martingale convergence theorem [39] and (C1)-(C2), (27) can be written as follows:

$$
\begin{aligned}
E[S_t | \mathscr{F}_{t-1}] &\leq S_{t-1} + \chi^2 \lambda_{max}[H_t \Gamma_{t,\Theta} H_t], \\
&0 < \chi_{min} < \chi < \chi_{max} < \infty,
\end{aligned}
\tag{28}
$$

where the conditional expectation is described by $E[\cdot|\cdot]$.

For further derivation, the function $T_t = \frac{S_t}{[\ln|H_t|]^\alpha}, \alpha > 1$ is given. Because $\ln|H_t|$ is non-decrease, (28) is transformed into the following form:

$$E[T_t | \mathscr{F}_{t-1}] \leq \frac{S_{t-1}}{[\ln|H_t|]^\alpha} + \frac{\chi^2 \lambda_{max}[H_t \Gamma_{t,\Theta} H_t]}{[\ln|H_t|]^\alpha}. \tag{29}$$

Since $\sum_{t=1}^{\infty} \frac{\chi^2 \lambda_{max}[H_t \Gamma_{t,\Theta} H_t]}{[\ln|H_t|]^\alpha}$ is finite, using the martingale convergence theorem to (29), and we can obtain that $T_t$ is convergent, i.e., $T_t$ converges to a finite random variable $T_0$,

$$T_t = \frac{S_t}{[\ln|H_t|]^\alpha} \to T_0 < \infty, a.s., \tag{30}$$

or

$$S_t = O([[\ln|H_t|]^\alpha]), a.s.. \tag{31}$$

According to the definition of $S_t$, we can obtain

$$\|\tilde{\Theta}_t\|^2 \le \frac{S_t}{\lambda_{min}[\Gamma_{t,\Theta}^{-1}]} \le O\left(\frac{[\ln|H_t|]^\alpha}{\lambda_{min}[\Gamma_{t,\Theta}^{-1}]}\right). \tag{32}$$

At this point, the proof process of Theorem 1 is completed.

## 5. Numerical example and experiment

This section offers the simulation and experiment for the designed scheme, which is also compared with other algorithms.

### 5.1 Numerical example

The linear subsystems $L_1$ and $L_2$ of sandwich system are given as follows:

$L_1$: $x_t = a_1 u_{t-1} + a_2 u_{t-2} - b_1 x_{t-1} - b_2 x_{t-2}$,

$L_2$: $y_t = c_1 v_{t-1} + c_2 v_{t-2} - d_1 y_{t-1} - d_2 y_{t-2} + w_t$.

The corresponding parameters are set to $a_1 = 1$, $a_2 = 0.35$, $b_1 = 0.5$, $b_2 = 0.45$, $c_1 = 1$, $c_2 = 0.1$, $d_1 = 0.4$, $d_2 = 0.3$, the parameters of deadzone are chosen as $k_L = 0.4$, $d_L = 0.1$, $k_R = 0.4$, $d_R = 0.1$. In this section, these parameters are estimated by using the developed approach and some comparison algorithms.

The efficiency of the developed estimation scheme in Section 3 for the sandwich system is studied on the simulation example. The virtue of the proposed method is checked by comparing the following identification approaches such as AM-RLS algorithm in [40], PPPE algorithm with linear filter in [41], and VRAE method with low pass filter in [42]. The input signal $u_t$ is a random signal, where its mean is zero, and its variance is one. The noise signal $w_t$ is a white noise. The main initial values of the considered identification schemes are listed as follows:

(1) AM-RLS algorithm: the initial values of the auxiliary model are

$0.001, \hat{\theta}(0) = [0.01, 0.001, 0.001, 0.001, 0.01, 0.001, 0.001, 0.001, 0.01, 0.01]^T$, $P(0) = 10^6 I$.

(2) PPPE with linear filter: $k = 7$, $l = 10$, $\Gamma(0) = 10 * diag([88.1, 31.5, 44.8, 40.6, 6.6, 90.2, 8.4, 21.2, 87.2, 70])$, $\hat{\theta}(0) = [0.01, 0.001, 0.001, 0.001, 0.01, 0.001, 0.001, 0.001, 0.01, 0.01]^T$.

(3) VRAE method with low pass filter: $f = 5$, $\eta = 0.1$, $\gamma = 1$,

$\hat{\theta}(0) = [0.01, 0.001, 0.001, 0.001, 0.01, 0.001, 0.001, 0.001, 0.01, 0.01]^T$, $\Gamma(0) = 10 * diag([195, 69.5, 98, 89, 14, 198, 18.2, 48.5, 192, 150])$.

(4) Proposed scheme: $v(0) = 7$, $\varrho_\infty$ 0.01, $\gamma = 8$, $\underline{\zeta} = 0.05$, $\bar{\zeta} = 0.5$,

$\hat{\theta}(0) = [0.01, 0.01, 0.01, 0.01, 0.01, 0.01, 0.01, 0.01, 0.01, 0.01]^T$, $\Gamma(0) = diag([5.3, 0.01, 0.01, 0.01, 0.01, 5.3, 0.01, 1.31, 5.41, 4])$, $\kappa(0) = 5$, $\varrho_0 = 0.5$.

In Figs 5–7, we show the evolution curves of the parameter estimation gained by the four estimators. From Figs 5–7, although the four algorithms can effectively identify the system parameters, the developed scheme in Section 3 performs an excellent performance in terms of convergence speed. Fig 8 shows that the estimation error by the presented scheme is constrained within the PCT predefined domain, which is designed according to the user's

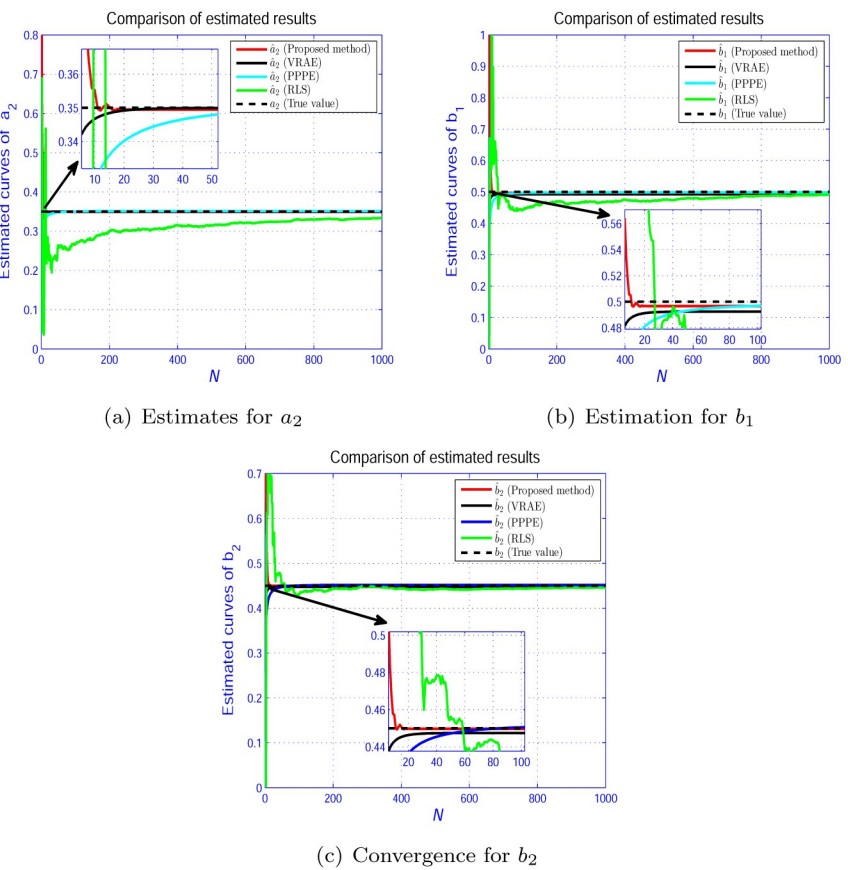

(a) Estimates for $a_2$

(b) Estimation for $b_1$

(c) Convergence for $b_2$

**Fig 5. Parameter estimation for $L_1$.** (a) Estimates for $a_2$. (b) Estimation for $b_1$. (c) Convergence for $b_2$.

requirement. The three other estimation approaches are outside the PPF bounds. This condition is because the constraint condition in (18) is considered.

Fig 9 plots the estimation results of the Monte Carlo run by producing 100 independent testing for the proposed method. As we can notice from Fig 9(a), the estimated parameters fluctuates near the desired value with the increase of the independent testing. The boxplot observed in Fig 9(b) displays that the estimated parameters give a higher concentrated distribution results.

To test the robustness of the developed scheme, the different noise intensities such as $\sigma^2 = 0.1^2, \sigma^2 = 0.5^2, \sigma^2 = 1^2, \sigma^2 = 2^2$, and $\sigma^2 = 5^2$ is injected in the system. The estimation error profiles with different noise are displayed in Fig 10. one may find that the estimation error produced by low noise (such as $\sigma^2 = 0.1^2$ and $\sigma^2 = 0.5^2$) is closer to the middle region of the prescribed boundary, while the estimation error caused by high noise (such as $\sigma^2 = 1^2$ and $\sigma^2 = 2^2$) intensity is close to the predefined boundary, but not beyond the preset area. Above results demonstrate that the proposed method has better robustness performance. Further increasing the intensity of noise (such as $\sigma^2 = 5^2$) will lead to the estimation error exceeding the preset boundary. Such phenomenon indicates that the high noise may extend the error boundary slightly in instantaneous performance convergence stage, one solution is to retune the predefined boundary in (17) with larger initial parameter and ultimate error value, such that the estimation error caused by high-intensity noise can also tend to be within the preset area.

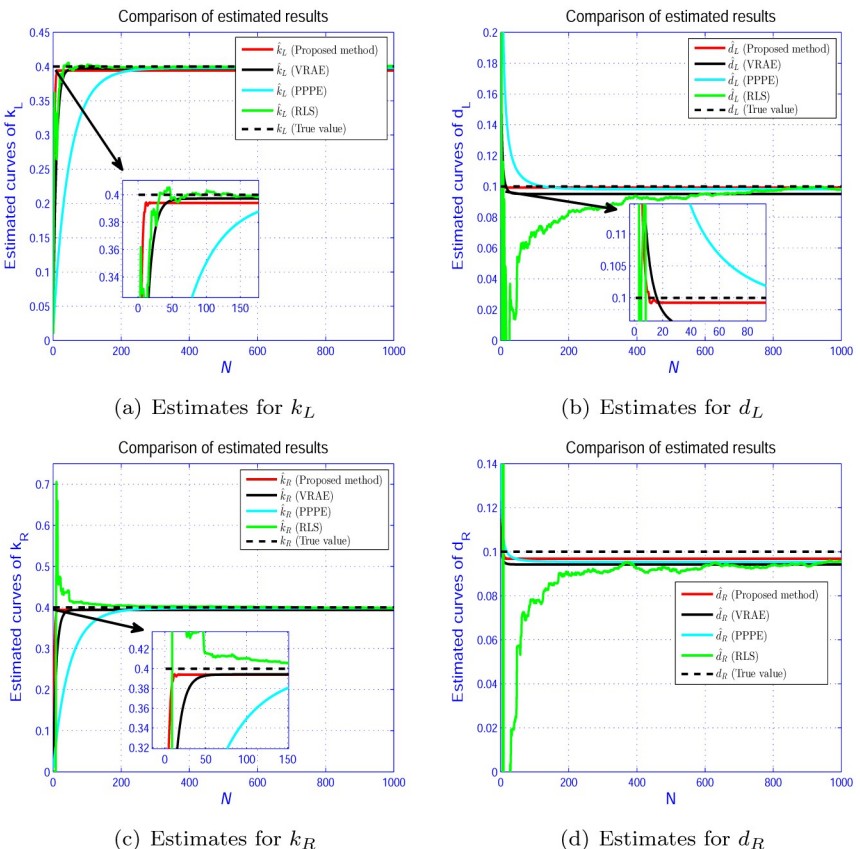

**Fig 6. Parameter estimation for deadzone.** (a) Estimates for $k_L$. (b) Estimates for $d_L$. (c) Estimates for $k_R$. (d) Estimates for $d_R$.

## 5.2 Experiment

The application of the proposed scheme is justified by considering the identification of the servo mechanism, as displayed in Fig 11. The experimental bed consists of a synchronous motor, an encoder, a power amplifier, a DSP and a stabilized platform. The result is displayed in PC with CCS3.0, $y_r = 0.8\sin(2/5\pi t)$ is fed into the system, and the sampling rate is set as 0.01 second.

The system is modeled as

$$\begin{cases} J\ddot{q} + T_f + T_l = T_m \\ T_m = K_T I_a \\ K_E \dot{q} + L_a \dfrac{dI_a}{dt} + R_a I_a = u, \end{cases}$$

where $J$ is the motor inertia, the friction is represented by $T_f$, the load is described by $T_l$, and $T_m$ is the torque. The electromechanical time and back-electromotive constants are denoted by $K_T$ and $K_E$, respectively. The resistance, inductance and current are denoted by $R_a$, $L_a$ and $I_a$, respectively. The angular position is denoted by $q$, the velocity is represented by $\dot{q}$.

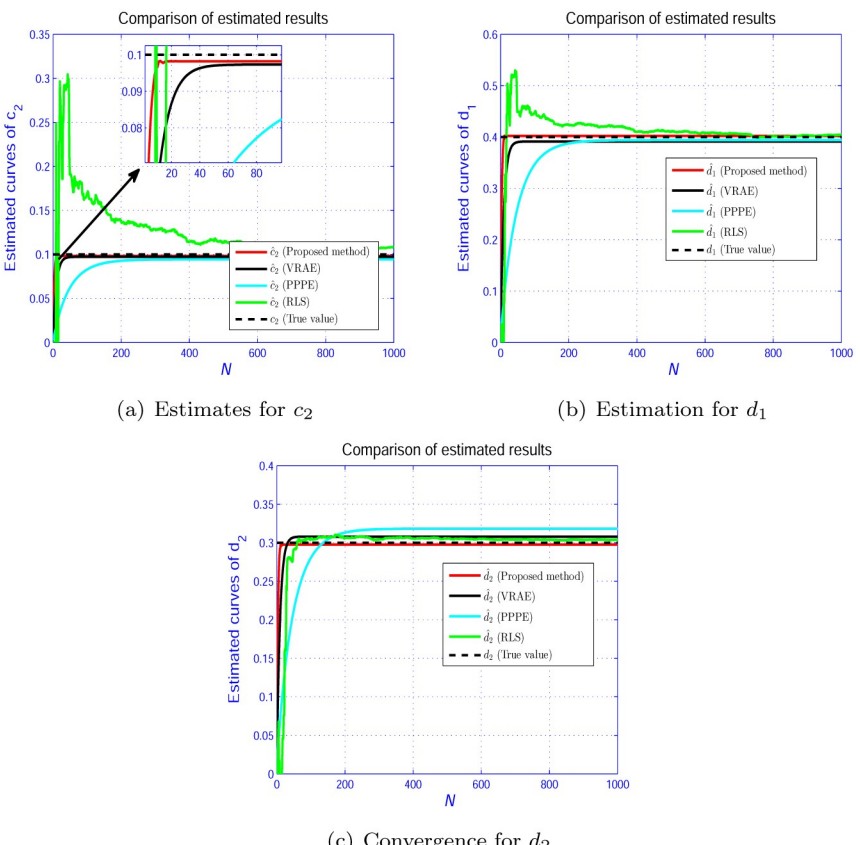

(a) Estimates for $c_2$    (b) Estimation for $d_1$

(c) Convergence for $d_2$

**Fig 7. Parameter estimation for $L_2$.** (a) Estimates for $c_2$. (b) Estimation for $d_1$. (c) Convergence for $d_2$.

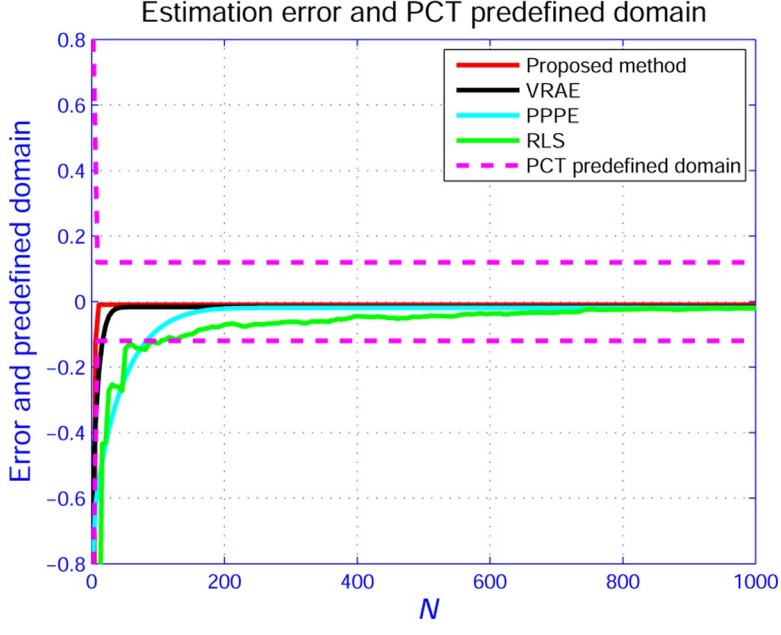

**Fig 8. Estimation error curves.**

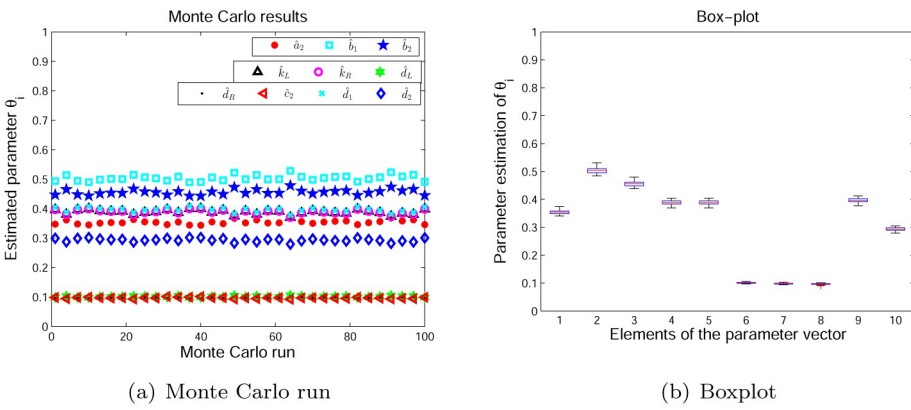

**Fig 9. Monte Carlo run.** (a) Monte Carlo run. (b) Boxplot.

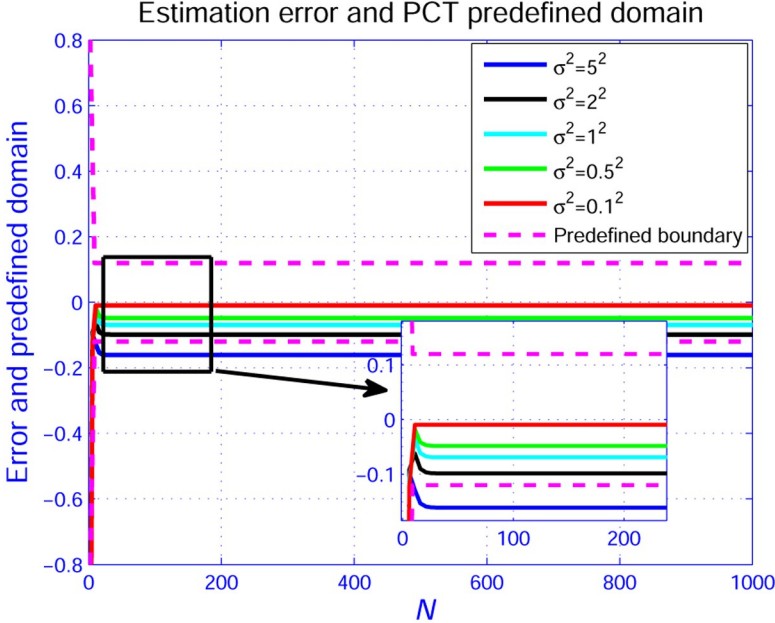

**Fig 10. Estimation error curves.**

Based on the above equation and defining $x = [x_1, x_2]^T = [q, \dot{q}]^T$ [43], the state equation form of system is written by

$$
\begin{cases}
\dot{x}_1 = x_2 \\
\dot{x}_2 = \dfrac{1}{J}(-K_2 x_2 + K_1 u - T_f - T_l),
\end{cases}
$$

or

$$
\dot{x}_2 = \varphi^T(t)\theta,
$$

where $K_2 = K_T K_E / R_a$, $T_f = T_c sgn(x_2) + B x_2$, $K_1 = K_T / R_a$.

The estimated values $\theta_1 = K_2/J$, $\theta_2 = K_1/J$, $\theta_3 = T_c/J$ and $\theta_4 = B/J$ are defined, the trajectories of the estimated value are plotted in Fig 12. we notice that parameter estimation by the

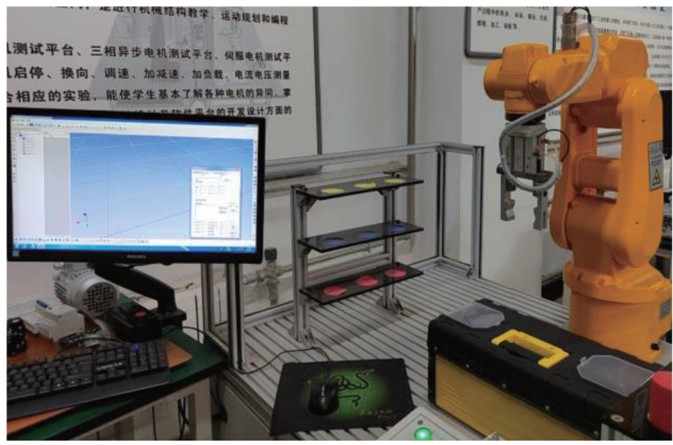

**Fig 11. Servo system.**

(a) Estimates for $\theta_1$

(b) Estimates for $\theta_2$

(c) Estimates for $\theta_3$

(d) Estimates for $\theta_4$

**Fig 12. Estimation histories of the system parameters.** (a) Estimates for $\theta_1$. (b) Estimates for $\theta_2$. (C) Estimates for $\theta_3$. (d) Estimates for $\theta_4$.

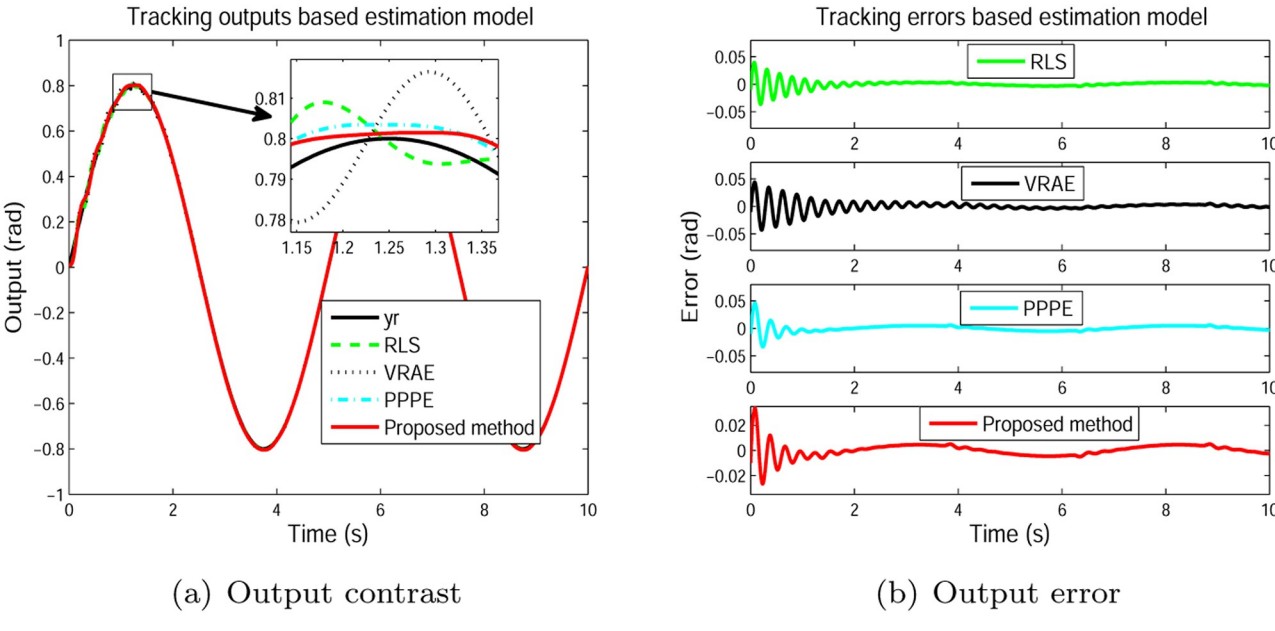

**Fig 13. Model output.** (a) Output contrast. (b) Output error.

considered estimation approaches can tend to the steady value over time. The RLS scheme has a vibration due to the lack of the filter, the PPPE method and VRAE scheme provide a smooth profile thanks to the filter, but PPPE gives a slow convergence rate. In comparison with the PPPE method, the convergence rate of the VRAE algortihtm is increased because of the variable gain. The presented approach offers faster rate than the other comparison algorithms because the error is restricted on the basis of the condition (18). Moreover, the proposed scheme has no overshoot. The estimation results indicate that the designed algorithm has excellent transient performance.

In accordance with the parameter estimation, the predictive outputs together with the desired output are presented in Fig 13. And it is easy to observe from Fig 13, the developed method gives more tracking output of the system with smaller tracking error comparing to the RLS, PPPE and VRAE algorithms. The tracking results indicate the advantage of the introduced estimation algorithm.

Three frequently-used performance indices, namely, mean of error (*Me*), normalised mean squared error (*Mse*) and root mean square of error (*Rmse*) [44] are given for the four different estimators to quantitatively illustrate the identification performance of the presented estimator. The values of contrastive indices for the four estimators are indicated in Table 1. It can be observed from Table 1 that the presented estimator provides the smallest values among the four considered indices, showing that the developed method offers better tracking

**Table 1. Quantitatively indices for experiment.**

| algorithm | *Me* | *Mse* | *Rmse* |
|---|---|---|---|
| RLS | $5.634 \times 10^{-1}$ | $7.389 \times 10^{-2}$ | $2.035 \times 10^{-1}$ |
| PPPE | $3.568 \times 10^{-1}$ | $6.721 \times 10^{-2}$ | $2.126 \times 10^{-1}$ |
| VRAE | $2.015 \times 10^{-1}$ | $4.358 \times 10^{-2}$ | $1.987 \times 10^{-1}$ |
| Presented scheme | $1.568 \times 10^{-1}$ | $5.241 \times 10^{-2}$ | $1.045 \times 10^{-1}$ |

performance over the three other identification schemes. This also reflects that the proposed method gives superior identification nature than the other given estimators.

## 6. Conclusion

This study proposes a new identification design for the sandwich system with deadzone non-linearity by using the PCT technique. Unlike the conventional reports, the developed estimator considers the instantaneous performance of the parameter identification. This identification method provides strong robustness ability by using an adaptive filter and lifts the utilization of data based on the variable fading factor. By proposing a novel adaptive law, the predefined domain on the identification error and convergence of the transformed error can be achieved. The convergence of the algorithm is rigorously proven through the usage of martingale convergence theorem. The illustrative example, practical application and performance indexes results are given to test the usefulness and practicality of the developed robust instantaneous performance estimator.

## Supporting information

**S1 Data.**
(ZIP)

## Author Contributions

**Conceptualization:** Lijun Ma.

**Data curation:** Yongqiang Wang.

**Funding acquisition:** Zhengbin Li.

**Investigation:** Lijun Ma.

**Methodology:** Zhengbin Li.

**Software:** Zhengbin Li.

**Validation:** Yongqiang Wang.

**Writing – review & editing:** Zhengbin Li.

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
