## [Decision Letter · Decision Letter 0]

20 Apr 2022

PONE-D-22-08046Parameter estimation for nonlinear sandwich system using instantaneous performance principlePLOS ONE

Dear Dr. Li,

Thank you for submitting your manuscript to PLOS ONE. After careful consideration, we feel that it has merit but does not fully meet PLOS ONE’s publication criteria as it currently stands. Therefore, we invite you to submit a revised version of the manuscript that addresses the points raised during the review process.

ACADEMIC EDITOR: Please consider all the comments from reviewers and carefully proof read the revised manuscript before re-submitting it.

We look forward to receiving your revised manuscript.

Kind regards,

Qichun Zhang, PhD

Academic Editor

PLOS ONE

Journal Requirements:

"This paper is supported by the Key Specialized Research and Development Projects of Henan Province under Grant 202102210337"

"NO"

5. Thank you for stating the following in the Financial Disclosure section: 

"NO"

We note that one or more of the authors are employed by a commercial company: Anyang Iron \\& Steel Co. Ltd.

Additional Editor Comments :

This paper investigates an interesting topic and would generate impact in practice. Basically, the writing needs to be improved and the typos should be corrected by a careful proof reading. Technically speaking, the contribution should be highlighted in revised version and the the details of technical design should be further explained as the comments mentioned that some assumptions cannot be met in real industrial processes. Therefore, a major revision is the decision for the current manuscript.

Reviewers' comments:

Reviewer's Responses to Questions

**Comments to the Author**

1. Is the manuscript technically sound, and do the data support the conclusions?

Reviewer #1: Yes

Reviewer #2: Yes

Reviewer #3: Yes

2. Has the statistical analysis been performed appropriately and rigorously? 

Reviewer #1: Yes

Reviewer #2: Yes

Reviewer #3: Yes

3. Have the authors made all data underlying the findings in their manuscript fully available?

Reviewer #1: Yes

Reviewer #2: Yes

Reviewer #3: Yes

4. Is the manuscript presented in an intelligible fashion and written in standard English?

Reviewer #1: Yes

Reviewer #2: Yes

Reviewer #3: Yes

5. Review Comments to the Author

Reviewer #1: This paper proposes a predefined performance estimation approach using predefined constraint technology and high-effective filter for sandwich system. Few findings are reported for the instantaneous performance of parameter estimation because the instantaneous performance is difficult to quantify based on the design algorithm. A numerical simulation and real-life process are employed to demonstrate the effectiveness of the proposed novel estimator.

CMT.1: In Introduction, the specific reasons for the difficulty of transient performance research can be explained, which reflects the innovation of the article.

CMT.2: Is the degree information of two linear systems known or need to be re-estimated

CMT.3: In real industry, the sine signal of this paper is chosen as estimate the nonlinear process. For me this can cause issue as the input signal is not of sufficiently high order.

CMT.4: Piecewise deadzone function is described by the function (3) instead of the function based on (7)-(8).

CMT.5: Compared with the classic PPF, the function proposed (see, (17)) in this paper is designed as a whole design plan.

CMT.6: Is the estimation error used by authors the difference between the real value and the estimated value, or other definition forms.

Reviewer #2: In this work，an instantaneous performance scheme of parameter estimation is introduced for nonlinear sandwich system. The content of the paper meets the requirements of the journal and the writing is standard. Some other comments have been uploaded as the attachment.

Reviewer #3: Comment

The almost existing identification algorithms do not consider the prescribed error bound on the parameter estimation error information, which may result in the poor transient performance of parameter estimation during the identification process. In this paper, a novel identification scheme is presented for nonlinear sandwich model, which is implemented by using prescribed performance function and error transformation technique. The numerical example and experiment results validate that the proposed scheme can provide more accurate parameter estimation and better transient performance than the existing methods.

Comment 1: If the noise attribute is unknown, how to design the de-noising filter of this paper.

Comment 2: There exist two linear subsystems in this paper, the assumptions on the two linear systems should be stated as formal assumptions.

Comment 3: In Eq.(15), please remove the necessary derivation steps and give the results directly, because these formulas are easy to deduce.

Comment 4: The presentation and language need be improved.

Comment 5: The auxiliary model method can indeed solve the problems in this paper, but the specific solution block diagram needs to be provided.

Comment 6: In the process of proving the convergence of the proposed algorithm, I am interested in the meaning of the symbolic representation of “O” in Eq.(31).

6. PLOS authors have the option to publish the peer review history of their article (what does this mean?). If published, this will include your full peer review and any attached files.

Reviewer #1: No

Reviewer #2: No

Reviewer #3: No

---

## [Author Response · Author response to Decision Letter 0]

12 Jun 2022

Reference No.: PONE-D-22-08046 R1

Title: “Parameter estimation for nonlinear sandwich system using instantaneous performance principle”

Dear Editor

Thank you for your mail dated Apr. 20 informing us further modification of the above paper. Following the provided insightful and valuable suggestions, the paper has been duly revised to deal with the concerned issues. Finally, we would again like to thank the editors and the reviewers for your supports and the time made in the reviewing process.

Yours sincerely,

Prof. Zhengbin Li

The major changes made in the revision were all marked in a red color in the revised manuscript.

Responses to Reviewer’ Comments:

Response to Reviewer 1

The authors would like to thank the reviewer for your support and comments in helping us to improve the paper. 

Comment 1: In Introduction, the specific reasons for the difficulty of transient performance research can be explained, which reflects the innovation of the article.

Response:

This reviewer is insightful. The specific reasons for the difficulty of transient performance research have been added in Introduction.

Comment 2: Is the degree information of two linear systems known or need to be re-estimated.

Response:

Thank you for this insightful comment. The degrees information of two linear systems are assumed to be known. According to suggestion, the degree information of two linear systems has been added to Assumption 1.

Comment 3: In real industry, the sine signal of this paper is chosen as estimate the nonlinear process. For me this can cause issue as the input signal is not of sufficiently high order.

Response:

Thank you for comment. 

In parameter estimation field, an input signal denotes a persistently exciting (PE) condition of order when (1) holds and (2) is a positive definite matrix [1]. 

 (1)

 (2)

In this paper, the input signal is chonsen as , by combining (1)-(2), if , it yields

 , 

due to the fact the sequential principal sub formula of are and ,

thus, (2) is a positive definite. 

When , we have .Therefore, if , the third-order system is not excited.

For the system of this paper, this system is second-order system. Thus, based on sine signal, the system feature can be fully excited.

Reference:

[1] Åström K. J., Wittenmark B. Adaptive control (2th edition) [M]. Prentice Hall, 1994.

Comment 4: Piecewise deadzone function is described by the function (3) instead of the function based on (7)-(8).

Response:

Thank you for this insightful suggestion. By using the functions (7)-(8), we can obtain the expression of the deadzone, its input-output relationship is the same as that of (3). The above relationship can be found in [1].

Reference:

[1]Vörös J . Iterative algorithm for parameter identification of Hammerstein systems with two-segment nonlinearities[J]. IEEE Transactions on Automatic Control, 1999, 44(11):2145-2149.

Comment 5: Compared with the classic PPF, the function proposed (see, (17)) in this paper is designed as a whole design plan.

Response:

Thank you for this insightful comment. The piecewise function PPF has singular value problem[32], to avoide such issue, the whole design plan is given, as shown in (17). 

Comment 6: Is the estimation error used by authors the difference between the real value and the estimated value, or other definition forms.

Response:

Thank you for this insightful comment. Your are right, the estimation error is defiend by using the difference between the real value and the estimated value in this paper. 

In addition, some literatures also use the percentage of parameter error to describe the result of estimation error [12,18].

Response to Reviewer 2

The authors would like to thank the reviewer for your support and comments in helping us to improve the paper. 

Comment 1: In Assumption 1, the author provides some assumptions, but does not offer the role of these assumptions.

Response:

Thank you for this valuable comment. Based on your suggestion, the explanations for these assumptions have been added to Assumption 1.

Comment 2:. The author proposes the estimation algorithm containing some parameters, and some parameter selection criteria can be added, so as to increase the integrity of the original manuscript.

Response:

Thank you for this valuable comment. Based on your suggestion, the parameter selection criteria have been added.

Comment 3:. As we all know, the forgetting factor can increase data utilization. Please clarify the advantages of variable fading factor in this article.

Response:

Thank you for this valuable comment. The adaptive forgetting factor proposed in this paper provides a large forgetting coefficient at the initial stage of parameter estimation, and processes a small forgetting coefficient at the later stage of parameter estimation. Therefore, it can avoid data saturation and improve data utilization, as shown in Remark 1.

Comment 4:. The filter 𝑣 is chosen upon the cutoff frequency. In engineering application, 100 Hz is selected in general. Then, 50 Hz is chosen as cutoff frequency. Is it better to use 𝑣 based on the sampling frequency?

Response:

Thank you for this valuable comment. In the application example, when the frequency of the acquired data is 2.56~4 times of the signal maximum frequency, the raw signal can be recovered using sampled digital signal [1]. 

Reference:

[1] Peiqing Cheng. Digital Signals Processing [M], Tsinghua university press,2015.

Comment 5:. In Experiment section, the identification model for servo system should be added to increase the readability of the study.

Response:

Thank you for this valuable comment. According to your comment, the identification model has been added in Experiment section.

Response to Reviewer 3

The authors would like to thank the reviewer for your support and comments in helping us to improve the paper. 

Comment 1: If the noise attribute is unknown, how to design the de-noising filter of this paper.

Response:

Thank you for this comment. In parameter estimation field，it has been shown that the prefiltering process is applied to obtain data polishing by using removing undesired disturbance features in the identification data when the noise is unknown or known. This is implemented primarily by filter from the noise data, and linear filter is common practice in applications of identification techniques [1]. 

In this paper, the filter operator is a linear filter, some criteria such as best transfer function estimation principle [1], prefiltered prediction error principle [2] and estimation error principle [3] can be applied to select the parameter value . In this paper, we use estimation error principle to select the parameters of the filter [3].

References:

[1] Wahlberg B., Ljung L. Design variables for bias distribution in transfer function estimation [J]. IEEE Transactions on Automatic Control, 1986, 31(2): 134-144.

[2] Rivera D.E, Pollard I.F., Garcia C E. Control-relevant prefiltering: A systematic design approach and case study [J]. IEEE Transactions on automatic control, 1992, 37(7): 964-974.

[3] Wang Y, Ding F. Novel data filtering based parameter identification for multiple-input multiple-output systems using the auxiliary model [J]. Automatica, 2016, 71: 308-313.

Comment 2: There exist two linear subsystems in this paper, the assumptions on the two linear systems should be stated as formal assumptions.

Response:

Thank you for this comment. The assumption for linear subsystems has been added in Assumption 1.

Comment 3: In Eq.(15), please remove the necessary derivation steps and give the results directly, because these formulas are easy to deduce.

Response:

Thank you for comment. Based on your suggestion, the result directly of the derivation steps in (15) has been revised.

Comment 4: The presentation and language need be improved.

Response:

Thank you for this comment. Based on your suggestion, the presentation and language have been revised.

Comment 5: The auxiliary model method can indeed solve the problems in this paper, but the specific solution block diagram needs to be provided.

Response:

Thank you for this comment. Based on your suggestion, the specific solution block diagram for auxiliary model has been added in the end of the Section 3.

Comment 6: In the process of proving the convergence of the proposed algorithm, I am interested in the meaning of the symbolic representation of “O” in Eq.(31).

Response:

Thank you for this comment. The symbolic “O” is a bounded quantity rather than a quantity of the same order.

---

## [Decision Letter · Decision Letter 1]

27 Jun 2022

Parameter estimation for nonlinear sandwich system using instantaneous performance principle

PONE-D-22-08046R1

Dear Dr. Li,

We’re pleased to inform you that your manuscript has been judged scientifically suitable for publication and will be formally accepted for publication once it meets all outstanding technical requirements.

Kind regards,

Qichun Zhang, PhD

Academic Editor

PLOS ONE

Additional Editor Comments (optional):

All the comments have been addressed well in the revised version. Therefore, the paper is recommended being accepted as it is.

Reviewers' comments:

Reviewer's Responses to Questions

**Comments to the Author**

1. If the authors have adequately addressed your comments raised in a previous round of review and you feel that this manuscript is now acceptable for publication, you may indicate that here to bypass the “Comments to the Author” section, enter your conflict of interest statement in the “Confidential to Editor” section, and submit your "Accept" recommendation.

Reviewer #1: All comments have been addressed

Reviewer #2: All comments have been addressed

Reviewer #3: All comments have been addressed

2. Is the manuscript technically sound, and do the data support the conclusions?

Reviewer #1: Yes

Reviewer #2: Yes

Reviewer #3: Yes

3. Has the statistical analysis been performed appropriately and rigorously? 

Reviewer #1: Yes

Reviewer #2: Yes

Reviewer #3: Yes

4. Have the authors made all data underlying the findings in their manuscript fully available?

Reviewer #1: Yes

Reviewer #2: Yes

Reviewer #3: Yes

5. Is the manuscript presented in an intelligible fashion and written in standard English?

Reviewer #1: Yes

Reviewer #2: Yes

Reviewer #3: Yes

6. Review Comments to the Author

Reviewer #1: I am satisfied with the revision, the technique issues are all addressed. The paper can be accepted in its presented form.

Reviewer #2: In this work，an instantaneous performance scheme of parameter estimation is introduced for nonlinear sandwich system. To achieve the above purpose, the estimation error information reflecting the transient performance of parameter estimation is procured using the developed some intermediate variables. Then, a predefined constraint function is used to prescribe the error convergence boundary, in which the convergence rate is lifted. The advantages and usefulness of the article are proved by examples. The major novelty of this paper is to use preset performance technology.

The revision is enough to be accepted for this journal. We have no other comments.

Reviewer #3: The author has dealt with all my opinions, and I am quite satisfied. I suggest the paper be accepted.

7. PLOS authors have the option to publish the peer review history of their article (what does this mean?). If published, this will include your full peer review and any attached files.

Reviewer #1: No

Reviewer #2: No

Reviewer #3: No

---

## [Editor Report · Acceptance letter]

11 Jul 2022

PONE-D-22-08046R1 

Parameter estimation for nonlinear sandwich system using instantaneous performance principle 

Dear Dr. Li:

I'm pleased to inform you that your manuscript has been deemed suitable for publication in PLOS ONE. Congratulations! Your manuscript is now with our production department. 

Kind regards, 

on behalf of

Dr. Qichun Zhang 

Academic Editor

PLOS ONE